



# Application of HIDRA2 Deep Learning Model for Sea Level Forecasting Along the Estonian Coast of the Baltic Sea

Amirhossein Barzandeh[1], Marko Rus[2,3], Matjaž Ličer[2,4], Ilja Maljutenko[1], Jüri Elken[1], Priidik Lagemaa[1], and Rivo Uiboupin[1]

[1]Department of Marine Systems, Tallinn University of Technology, Tallinn, Estonia
[2]Slovenian Environment Agency, Ljubljana, Slovenia
[3]Faculty of Computer and Information Science, Visual Cognitive Systems Lab, University of Ljubljana, Ljubljana, Slovenia
[4]National Institute of Biology, Marine Biology Station, Piran, Slovenia

**Correspondence:** Amirhossein Barzandeh (amirhossein.barzandeh@taltech.ee)

**Abstract.** Sea level predictions, typically derived from 3D hydrodynamic models, are computationally intensive and subject to uncertainties stemming from physical representation and inaccuracies in initial or boundary conditions. As a complementary alternative, data-driven machine learning models provide a computationally efficient solution with comparable accuracy. This study employs the deep learning model HIDRA2 to forecast hourly sea levels at five coastal stations along the Estonian coast-
line of the Baltic Sea, evaluating its performance across various forecast lead times. Compared to the regional NEMO$_\mathrm{BAL}$ and subregional NEMO$_\mathrm{EST}$ hydrodynamic models, HIDRA2 consistently delivers superior results, particularly across all sea level ranges and stations. While HIDRA2 struggles to capture high-frequency variability above $(6\ \mathrm{h})^{-1}$, it effectively reproduces energy in lower-frequency bands below $(18\ \mathrm{h})^{-1}$. Errors tend to average out over longer time windows encompassing multiple seiche periods, enabling HIDRA2 to surpass the overall performance of the NEMO models. These findings underscore
HIDRA2's potential as a robust, efficient, and reliable tool for operational sea level forecasting and coastal management in the Eastern Baltic Sea region.

## 1 Introduction

The importance of sea level variability has grown significantly in the context of climate change as higher global temperatures accelerate its complexity (Bindoff et al., 2007; Cazenave et al., 2014). The increasing frequency and severity of extreme
weather events, compounded by rising sea levels due to climate change, present pressing challenges for coastal communities worldwide. Accurate and timely forecasting of sea level is essential for effective disaster preparedness, risk mitigation, and coastal management. However, forecasting sea level variability is inherently complex and requires sophisticated models that account for various drivers and uncertainties. Traditionally, sea level forecasting has been accomplished using hydrodynamic models, but these models are subject to a range of uncertainties induced by errors in atmospheric and open boundary forcing
fields, as well as physical flux parametrization (Church et al., 2011; Miles et al., 2014; Khojasteh et al., 2021). Errors in regional sea level prediction can arise from inaccurate initial and boundary conditions, models' approximated descriptions of physical





environments (e.g., coupling with waves, atmosphere, ice, and runoff), and the nonlinear nature of planetary dynamics (Carson et al., 2019; Ponte et al., 2019; Hamlington et al., 2020).

Despite these challenges, advancements such as assimilation of observation data in hydrodynamic models have shown promise in improving the accuracy of sea level prediction (Bajo et al., 2019; Barron et al., 2004; Tanajura et al., 2015; Liu and Fu, 2018). However, the computational cost of both ocean modeling and data assimilation adds to the complexity and resource requirements of accurately predicting sea levels (Byrne et al., 2023; Bajo et al., 2023). In this regard, the use of data-driven models as a surrogate system can be a promising addition to classic hydrodynamic models for addressing these challenges (Sonnewald et al., 2021). In recent years, machine learning techniques have demonstrated significant potential in

enhancing the accuracy and reliability of predictive models in oceanography (Ahmad, 2019; Lou et al., 2023; Jahanmard et al., 2023). Recent studies have highlighted the effectiveness of deep learning methods in predicting sea level heights across diverse scenarios and basins (Rus et al., 2023; Balogun and Adebisi, 2021; Rajabi-Kiasari et al., 2023; Bahari et al., 2023). However, these studies imply that, unlike hydrodynamic models, deep-learning models require particular structural design and adjustments depending on various objectives, and the specific characteristics of the study area. Additionally, ensemble-

based approaches in deep learning, widely employed in practical applications, distinguish themselves from traditional machine learning methods by generating and integrating multiple hypotheses from training data to improve prediction accuracy. This allows for the incorporation of more advanced and diverse algorithms, making it better suited to tackle specific challenges in operational applications (Zhang et al., 2022; Ganaie et al., 2022).

    The Baltic Sea, a large, high-latitude, semi-enclosed, and nearly tideless sea located in northern Europe is a vital marine

region characterized by its unique geographical and ecological features. It comprises several sub-basins and is connected to the Atlantic Ocean through the narrow Danish Straits in the southwest. This study specifically focuses on the Eastern Baltic Sea, which spans from approximately 21.5°E to 30.5°E longitude and 56.5°N to 61°N latitude. This region encompasses the northeastern Baltic Proper, the Gulf of Finland, and the Gulf of Riga—areas of strategic importance due to the heavy vessel traffic between their populated shores and islands. Understanding sea level changes in the Eastern Baltic Sea is therefore vital.

Sea level fluctuations affect navigation routes, harbor operations, and maritime safety, making accurate knowledge of these variations essential for stakeholders to manage risks, optimize vessel operations, and ensure efficient trade flows.

    Research on sea level variations in the Baltic Sea has revealed a wide range of patterns and trends influenced by numerous environmental factors that vary both spatially and temporally (Andersson, 2002; Ekman, 2009; Hünicke and Zorita, 2016). Historical tide gauge records, used by Jevrejeva et al. (2003) to reconstruct past sea level changes, demonstrate significant

interannual and decadal variability attributed to atmospheric pressure fluctuations and shifting wind patterns. Furthermore, the intricate structure of the Estonian coastline, combined with its numerous small islands, poses additional challenges for projecting the impacts of climate change and sea-level rise in this region (Johansson et al., 2004; Kont et al., 2008). Although predictions of sea level height on longer temporal scales, such as monthly or weekly, tend to be more accurate due to the smoothing of short-term fluctuations caused by local wind tilt and storm surges (Samuelsson and Stigebrandt, 1996; Scotto

et al., 2009; Elken et al., 2024), short-term sea level changes in the Baltic Sea are heavily influenced by dynamic factors such as wind, precipitation, and the redistribution of water through the Danish Straits (Hünicke et al., 2015). Extreme sea levels





on the Baltic coasts, resulting from increasingly frequent storm surges, are influenced by three main factors: the volume of water in the Baltic Sea's basins, which sets the initial sea level; the direction, speed and duration of tangential wind stresses, whether shoreward or seaward; and the rapid passage of deep low-pressure systems that deform the sea surface, producing

seiche-like oscillations that further alter sea levels (Kulikov and Medvedev, 2013; Wolski et al., 2014; Weisse et al., 2021; Wolski and Wiśniewski, 2021; Rutgersson et al., 2022). Seiche oscillations are especially significant in the Baltic Sea due to their prolonged decay over several days. Both the Gulf of Riga and the Gulf of Finland can experience their own seiches with different characteristics (e.g., the 5-hour seiche period of the Gulf of Riga, which differs from the recognized seiche period in the Gulf of Finland). These seiches may interact with those from neighboring basins (e.g., the approximately 27-hour seiche

period of the entire Baltic Sea), leading to complex phase relationships and further complicating sea level predictions (Metzner et al., 2000; Suursaar et al., 2002; Jönsson et al., 2008). This intricate interplay of atmospheric, oceanic, and coastal dynamics makes the prediction of sea level in the Baltic Sea a challenging task.

In response to the increasing demand for accurate and efficient forecasting methods in coastal regions, this study explores the potential of a deep-learning ensemble model. By utilizing deep neural networks, we aim to improve sea level prediction

accuracy in the Eastern Baltic Sea by applying our approach to a selection of sample stations. The primary focus of this study is to evaluate the effectiveness of a deep-learning ensemble model HIDRA2 (Rus et al., 2023) for sea level forecasting in the study area in multiple lead time windows. A secondary objective is to compare its performance with that of regional and subregional hydrodynamic models currently used for operational purposes in the study area.

## 2 Data and Methods

### 75 2.1 Tide gauge data

The hourly sea surface height (SSH) time series were extracted and evaluated as measures of sea level using the available data from five observation stations: Narva-Jõesuu, Pritia, Haapsalu, Pärnu, and Roomassaare (Figure 1). These stations are situated along the Estonian coast, in the Gulf of Finland and the Gulf of Riga. The datasets were provided by the Estonian Environmental Agency (https://www.ilmateenistus.ee/).

### 80 2.2 HIDRA2: A deep-learning ensemble sea level forecasting model

HIDRA2 is a state-of-the-art deep-learning model for predicting SSH at specific geographic locations, presented in depth in Rus et al. (2023). The model utilizes historical and predicted atmospheric data, tidal information, and observed SSH measurements to generate precise hourly SSH predictions for a 72-hour window. Unlike models that forecast the residual SSH, i.e., the difference between SSH and tide, HIDRA2 predicts the full SSH. The HIDRA2 architecture, illustrated in Figure 2, incor-

porates specialized encoders for processing both atmospheric data and sea level inputs. The Atmospheric encoder processes wind and pressure sequences, subsequently merging them into a unified feature representation. The SSH encoder processes historical SSH data. In the Fusion-regression block, all extracted features with the preceding 72-hour raw SSH data are fused





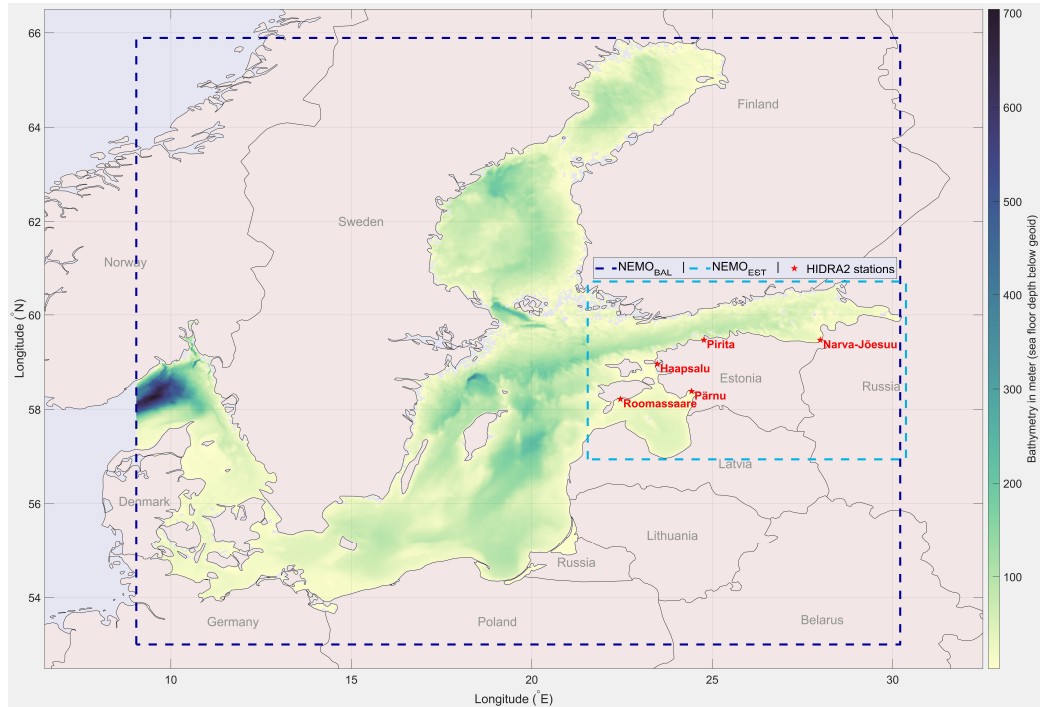

**Figure 1.** Study area: The domain of each model is illustrated, with red stars indicating the locations of the stations where HIDRA2 has been applied.

and regressed into final SSH forecasts. These encoders consist of a series of convolution layers, activation functions, dropout layers, and skip connections, forming a robust and deep neural network. Details of the encoding architecture are presented in

(Rus et al., 2023). The specific difference in this study for the Baltic Sea application is that, unlike in Rus et al. (2023), HIDRA2 was trained on the full SSH data rather than on separate tidal and SSH signals. This decision was made because tides in the Baltic are very weak, and the explicit inclusion of tides as a separate input signal did not improve forecasting performance. Consequently, we removed the tidal encoder block and fused only the atmospheric and SSH features, augmented by the raw SSH measurements, to produce the final set of 72-hour SSH predictions.

For the training of HIDRA2, we utilized atmospheric data fields from the European Centre for Medium-Range Weather Forecasts (ECMWF) Ensemble Prediction System (Leutbecher and Palmer, 2007). For all locations, we used 10-meter winds and mean sea level air pressure from a single atmospheric ensemble member, covering a longitudinal range from $16.25°E$ to $28.5°E$ and a latitudinal range from $54.25°N$ to $64°N$. The original meteorological data, with a domain size of $40 \times 50$, were subsampled to a $9 \times 12$ grid using linear interpolation to match HIDRA2's required input size. The training data for SSH at the

coastal stations were obtained from the Estonian Environmental Agency. It included hourly automated SSH observations from the period 2010 to 2019. Following the methodology outlined by Rus et al. (2023), we trained a separate HIDRA2 model for each of the five geographic locations depicted in Figure 1.





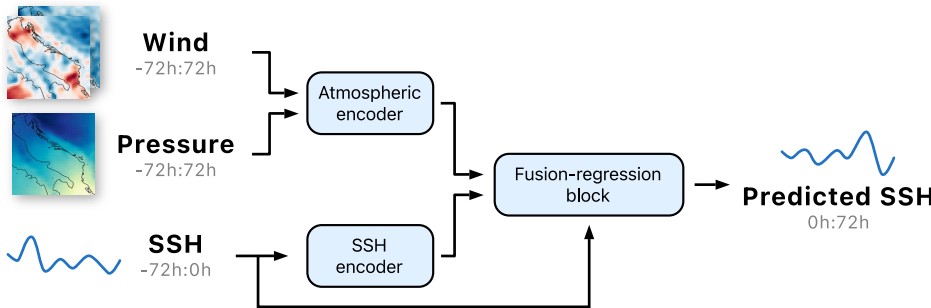

**Figure 2.** This diagram illustrates the HIDRA2 architecture.

## 2.3 NEMO$_{BAL}$: A forecast product based on a regional hydrodynamic modelling

We extracted the Baltic Sea Physics Analysis and Forecast datasets (BALTICSEA_ANALYSIS_FORECAST_PHY_003_006)

from the Copernicus Marine Service for each station in the study area over the study period. This forecast product is based on the NEMO 4.0 ocean engine, specifically configured for the North Sea and Baltic Sea as Nemo-Nordic 2.0 (Madec et al., 2017; Kärnä et al., 2021), and has been adapted and validated by the Baltic Monitoring and Forecasting Centre (BALMFC) to address the needs of the Baltic Sea region. For brevity, we refer to it as NEMO$_{BAL}$ in this study. The product offers a spatial resolution of 1 nautical mile and is updated twice daily, providing new six-day forecasts. Consequently, the dataset records available from

Copernicus include only 12-hour forecasts. The detailed validation report is given in Jandt-Scheelke et al. (2023).

## 2.4 NEMO$_{EST}$: A subregional hydrodynamic forecasting model

NEMO-EST05, is the Estonian adaptation of the Baltic Sea regional configuration Nemo-Nordic 2.0 with enhanced horizontal and vertical resolution, tailored for national operational purposes (hereafter referred to as NEMO$_{EST}$). Detailed specifications of this model are provided in Maljutenko et al. (2022), and an example of its utilization is presented in Pärt et al. (2023). NEMO$_{EST}$

uses the same NEMO 4.0 ocean engine with a horizontal resolution of 0.5 nautical mile spanning from 21.5°E to 30.5°E and from 56.5°N to 60.5°N. The meteorological forcing for the model system is obtained from the ECMWF 24-hour integrated forecasting system (Owens and Hewson, 2018). Boundary conditions for salinity, temperature, and sea levels are derived from the same NEMO$_{BAL}$, following the methodology outlined by (Elken et al., 2021). Model initialization in the October 2022 is from the NEMO$_{BAL}$. It can be inferred that NEMO$_{EST}$ serves as a down-scaled version of NEMO$_{BAL}$. Therefore, using both

models in the present study also provides implicit insights into how the spatial resolution of hydrodynamic models as well as their domain extent affects SSH forecasting.

## 2.5 Forecasting and evaluation framework

HIDRA2 and NEMO$_{EST}$ provide SSH forecasts for the next 72 hours, segmented into three 24-hour intervals: 0 to 24 hours (24), 24 to 48 hours (48), and 48 to 72 hours (72), with daily updates. This segmentation allows for the evaluation of model





performance at different lead times up to 72 hours. Regardless of the training time window used for the deep learning model, the testing period of all models in the present study spans one year, from April 1, 2023, to March 31, 2024 (in total, 8784 hourly time instances). This period aligns with the initiation of archiving 72-hour SSH fields in NEMO$_{\text{EST}}$ from its daily forecast cycles. For comparison, NEMO$_{\text{BAL}}$ was also considered as the publicly available SSH forecast for the study area. To ensure a fair comparison and address potential bias from the reference sea level in the hydrodynamic models (NEMO$_{\text{BAL}}$ and NEMO$_{\text{EST}}$), the outputs were adjusted by removing their mean value over the study period and then adding the mean observed SSH at each station. This adjustment to the datums of the local observation system ensures that the outputs of the hydrodynamic models are comparable to the observed values, whereas HIDRA2, as a data-driven model, is not subject to this type of reference sea level bias. Additionally, HIDRA2, as an ensemble forecasting system, produces 50 distinct forecast ensembles at each time point. In the analysis, unless specifically examining individual ensembles, the HIDRA2 forecasts are represented by the average of all 50 ensembles for each time instance and station. Using the mean of all ensembles as the final forecast is appropriate because the ensembles represent probabilities without any inherent priority in real-world occurrences.

## 2.6 Validation metrics

The validations and comparisons in the present study include the Pearson correlation coefficient (Correlation) and the Root Mean Square Deviation (RMSD). Each is defined as follows.

RMSD is given by Equation (1):

$$\text{RMSD} = \left( \frac{1}{n} \sum_{i=1}^{n} (o_i - m_i)^2 \right)^{1/2} \tag{1}$$

where $n$ represents the total number of observations, and $o_i$ and $m_i$ are the observed and modeled sea level values, respectively.

Correlation is defined by Equation (2):

$$\text{Correlation} = \frac{1}{\sigma_m \sigma_o} \sum_{i=1}^{n} (m_i - \overline{m})(o_i - \overline{o}) \tag{2}$$

where $\overline{m}$ and $\overline{o}$ are the mean values of the modeled and observed datasets, respectively.

The standard deviations of the modeled and observed datasets, $\sigma_m$ and $\sigma_o$, are calculated as shown in Equation (3):

$$\sigma_m = \left( \sum_{i=1}^{n} (m_i - \overline{m})^2 \right)^{1/2}, \quad \sigma_o = \left( \sum_{j=1}^{n} (o_j - \overline{o})^2 \right)^{1/2} \tag{3}$$

## 3 Results

### 3.1 Comparison of models' overall performance

Table 1 presents the RMSD values for the 12-hour, 24-hour, and 72-hour lead-time forecasts available during this study across all tide gauge stations. At the 24-hour lead time, the HIDRA2 ensemble mean consistently achieves the lowest RMSD at each



station, outperforming even NEMO$_{BAL}$, which inherently includes 12-hour lead-time forecasting. HIDRA2 with a 72-hour lead-time also demonstrates performance comparable to NEMO$_{BAL}$ at Roomassaare and Pärnu. Remarkably, HIDRA2 exhibits substantially lower RMSD at other stations, even with extended forecast lead times.

Figure 3 illustrates the forecast RMSD by sea level bin (0.05 m) for all models at different lead times. The RMSDs for each station are calculated using sea level data from April 2023 to April 2024. HIDRA2 with a 24-hour lead time generally outperformed the other models, achieving the lowest RMSD across most of the sea level distribution. However, in the tails of the distribution (SSH > 1.00 m or SSH < -0.40 m), RMSDs increased for all models. At extreme sea levels, HIDRA2 and NEMO$_{BAL}$ demonstrated similarly robust performance, albeit with differences depending on lead time. HIDRA2 showed

solid performance at the 24-hour lead time but struggled in the distribution tails at the 72-hour lead time. Figure 3 highlights that HIDRA2's advantage over NEMO$_{BAL}$ primarily stems from its superior performance in the bulk of the distribution and at extreme low SSH values. This distinction is particularly evident at the Haapsalu station.

**Table 1.** RMSD metrics of the three forecast models for the April 2023 to April 2024 period. Results are separated by the forecast lead times (12 h, 24 h, and 72 h). Lowest RMSDs are typed bold.

| Station | 12 h RMSD (m) | 24 h RMSD (m) | | 72 h RMSD (m) | |
| --- | --- | --- | --- | --- | --- |
| | NEMO$_{BAL}$ | NEMO$_{EST}$ | HIDRA2 | NEMO$_{EST}$ | HIDRA2 |
| Roomasaare | 0.051 | 0.058 | **0.033** | 0.063 | **0.051** |
| Pärnu | 0.074 | 0.074 | **0.051** | 0.083 | **0.074** |
| Haapsalu | 0.099 | 0.090 | **0.040** | 0.095 | **0.063** |
| Pritia | 0.056 | 0.059 | **0.039** | 0.067 | **0.054** |
| Narva-Jõesuu | 0.079 | 0.074 | **0.039** | 0.090 | **0.075** |

Additionally, the accuracy of each model, integrated across all study stations, is compared using Taylor diagrams (Taylor, 2001) in Figure 4a. The performances are also separately assessed during extreme negative and positive observed SSH events

(Figure 4 b & c). Extreme negative SSH values are defined as those falling below the 5th percentile of observed SSH during the study period at each station, while extreme positive values exceed the 95th percentile (Cannaby et al., 2016; Mentaschi et al., 2023). The 5th and 95th percentile thresholds for each station are provided in Table 2. HIDRA2 consistently outperformed both hydrodynamic models, showing the lowest RMSD and highest correlation coefficients across most observed SSH ranges. Although HIDRA2's accuracy declines for extreme SSH values, particularly at extended lead times, it still surpasses both

regional and subregional hydrodynamic models, except at the 72-hour lead time. Among the hydrodynamic models, the regional model NEMO$_{BAL}$ performs best during extreme SSH events, while the subregional model NEMO$_{EST}$ performs better under non-extreme high SSH conditions but struggles to accurately predict extreme SSH values.




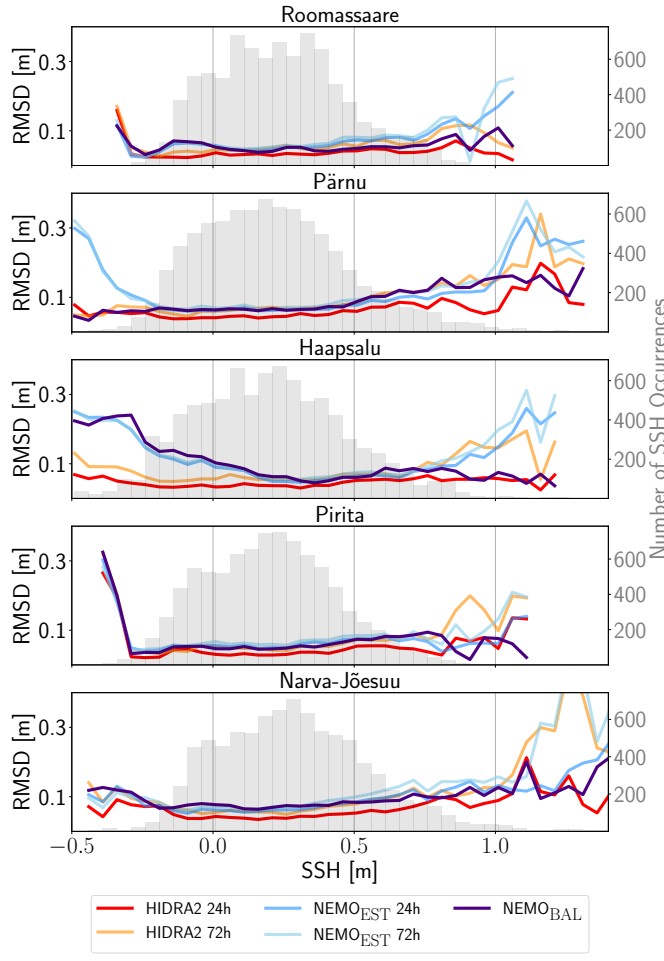

**Figure 3.** RMSD of all forecast models (colored lines, see legend at the bottom) at different sea level values. Grey histogram in the back (right axis) shows the distribution of occurences of a given sea level bin.

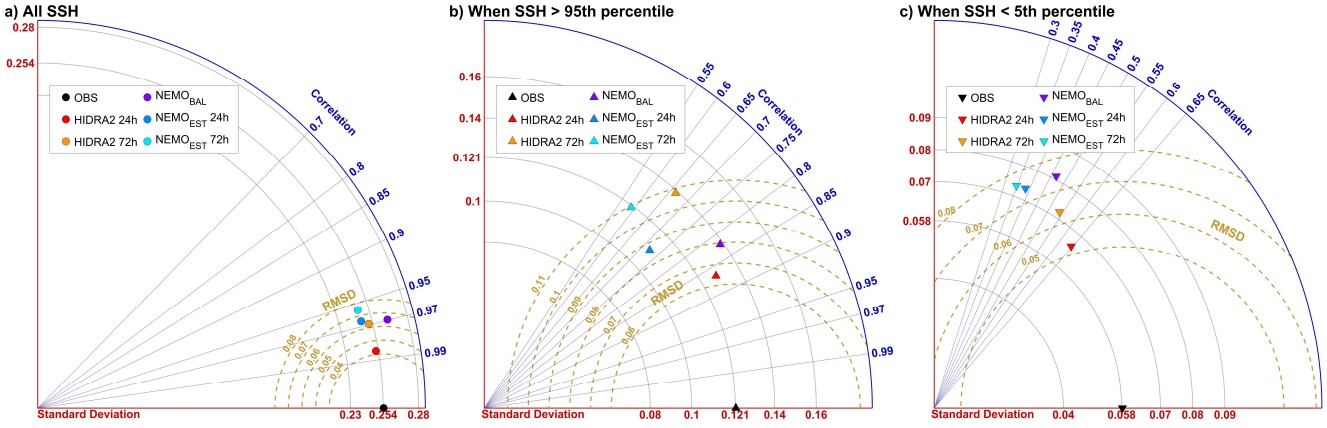

**Figure 4.** Taylor diagrams comparing forecasts: (a) all hourly time instances, (b) positive extreme SSH, and (c) negative extreme SSH.





**Table 2.** Thresholds for SSH used to identify extreme positive and negative storm surges at each station.

| Stations | $SSH_{05th-percentile}$ | $SSH_{95th-percentile}$ |
|---|---|---|
| Roomasaare | -0.11 | 0.67 |
| Pärnu | -0.16 | 0.71 |
| Haapsalu | -0.21 | 0.66 |
| Pritia | -0.15 | 0.60 |
| Narva-Jõesuu | -0.10 | 0.76 |

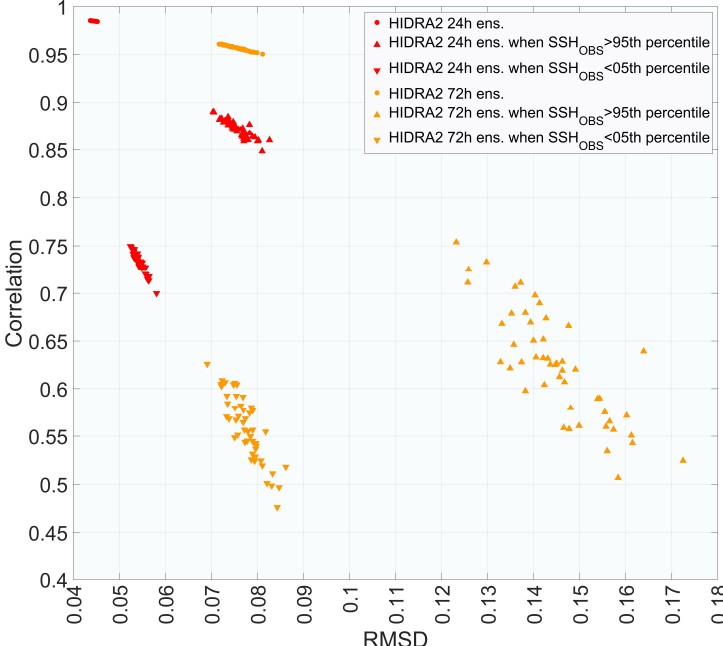

**Figure 5.** Performance of individual HIDRA2 ensemble members at 24-hour and 72-hour lead times and a further assessment under the conditions of the positive and negative extreme observed SSH.

We perform an independent analysis of the predictive accuracy across the 50 members of the HIDRA2 ensemble, evaluating each member's forecasts of SSH values. As expected, increasing the lead time of forecasts results in reduced accuracy of

each predeicted ensemble member. Both the correlation coefficient and RMSD vary across different lead times, and the spread of ensemble predicted values increases with longer lead times. This spread is especially pronounced during extreme observed SSH values. For negative extreme SSH events, correlation is lower with a lower RMSD, while for positive extreme SSH events, correlation is higher but with a higher RMSD.





## 3.2 Specific challenges in model performance

Figure 6 presents the predictions of HIDRA2 with a 24-hour lead time, alongside those of NEMO$_{BAL}$, for the entire study period at each station. For clarity, the performance of NEMO$_{EST}$, which is the least accurate (as detailed in Section 3.1), is not included in the figure. Both models exhibit stable behavior over the timescales considered.

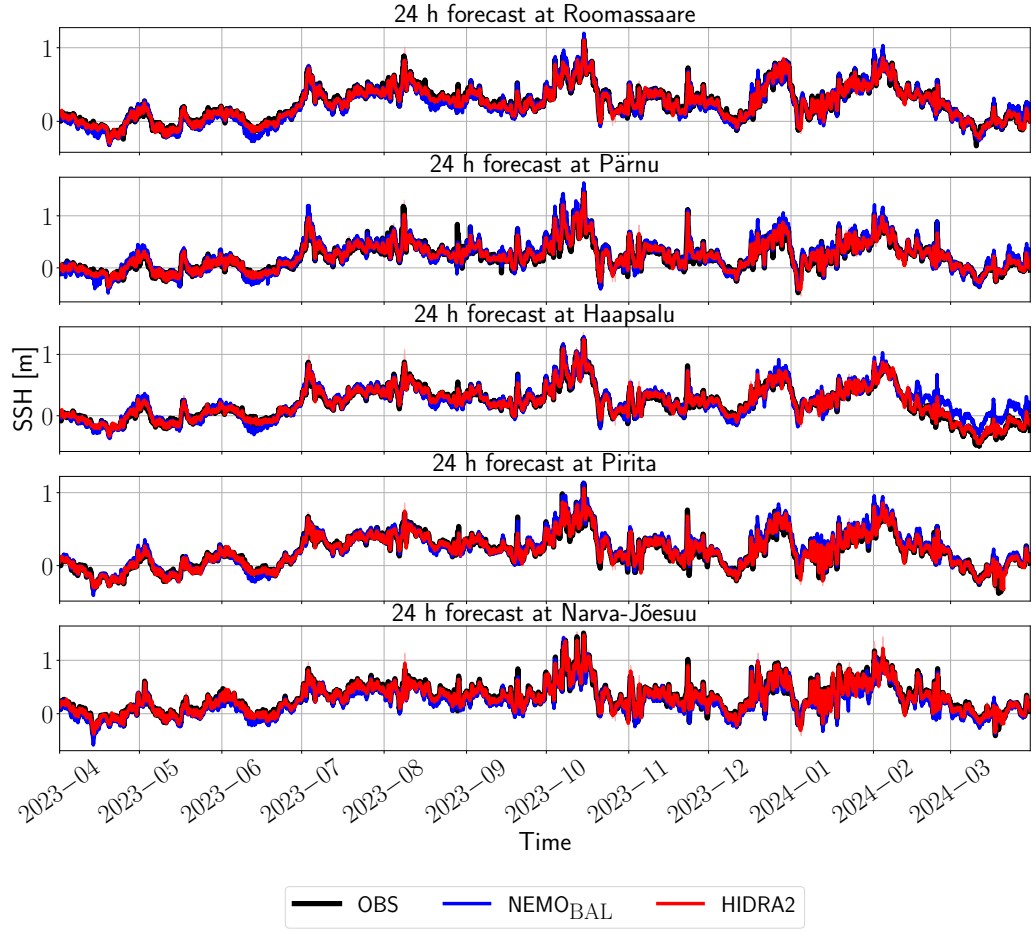

**Figure 6.** Observed SSH (black line) versus NEMO$_{BAL}$ (blue line) and HIDRA2 ensemble mean (red line) forecasts with a 24 h lead time at various stations between April 2023 and April 2024.

However, a closer examination, as shown in Figure 7, reveals that HIDRA2 predictions are overly smoothed, lacking the ability to fully capture high-frequency variability (with periods below 6 hours). In contrast, NEMO$_{BAL}$ demonstrates better
performance in this energy band. Figure 7 highlights model forecasts with a 24-hour lead time during October 2023, a period marked by a series of extreme sea-level events.



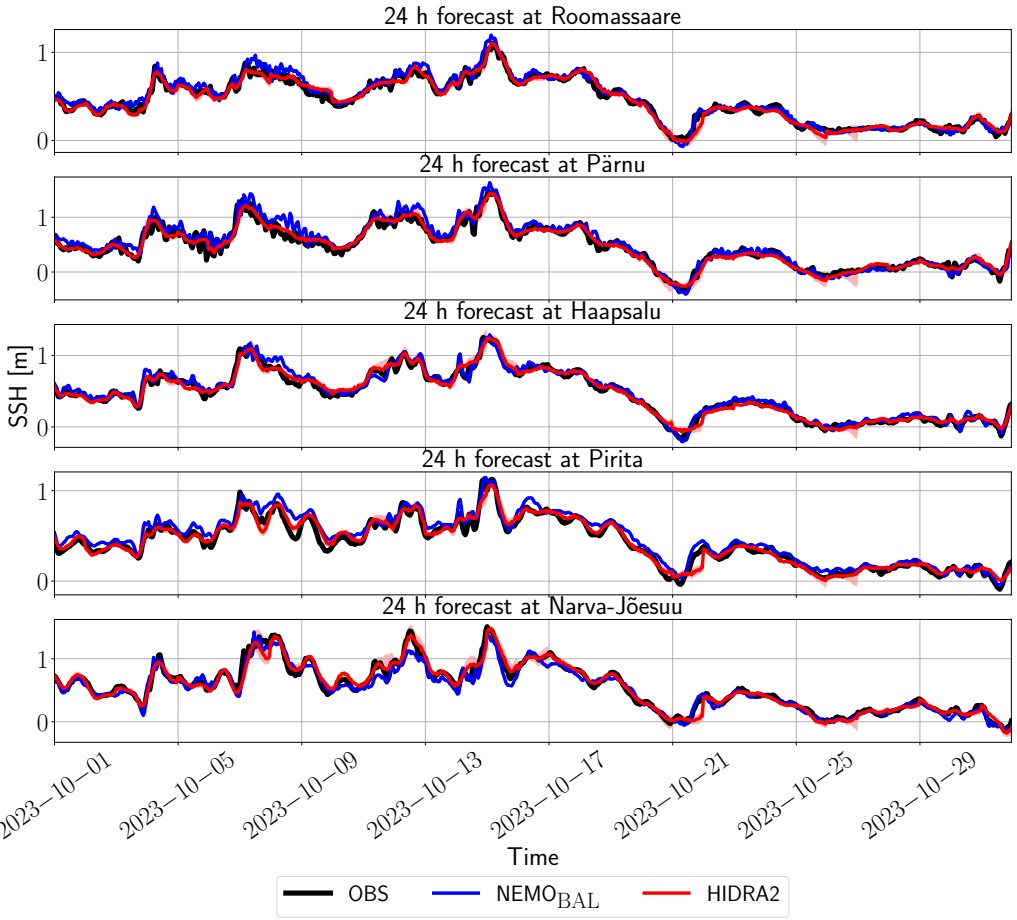

**Figure 7.** Same as Figure 6, but only for October 2023.

A visual comparison of model predictions on Figure 7 between 5 and 9 October 2023 is quite telling. Sea levels in Pärnu indicate the presence of the fundamental seiche with a period around 5 h. NEMO$_{\text{BAL}}$ is reproducing this excitation very clearly while it is practically completely absent from the HIDRA2 forecast. A similar scenario occurs on 13 October - we have high frequency variability in NEMO$_{\text{BAL}}$ and a completely smooth HIDRA2 ensemble mean. One might argue that this smoothing stems from the fact that we are working with the ensemble mean in which otherwise present oscillations in different ensemble members cancel out. This is however not the case. None of the ensemble members exhibit enough excitations in the Pärnu seiche energy band (not shown).

The limitations of HIDRA2 during seiche excitations can be evaluated by applying a band-pass filter to the sea-level signal in the Pärnu seiche energy band for the period of excited seiches between 5 and 9 October 2023. We use a 6th order Butterworth filter with a sample rate of 1 h and low and high cutoff frequencies of $(3\text{h})^{-1}$ and $(8\text{h})^{-1}$, respectively (Figure 8). A clear excitation of the seiche is visible in filtered observations in Figure 8 from 3 October on NEMO$_{\text{BAL}}$ captures this excitation





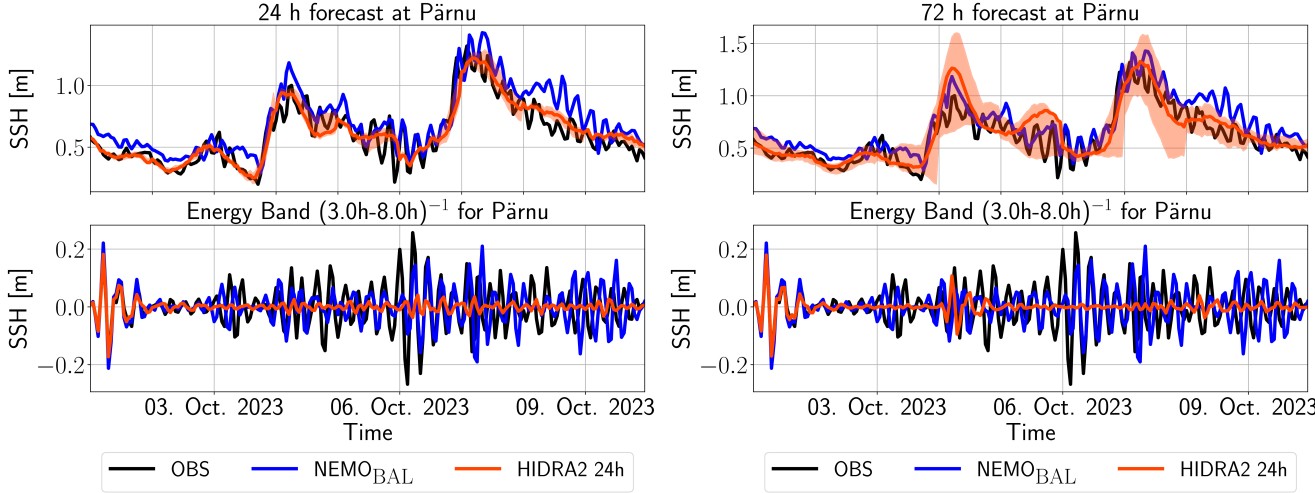

**Figure 8. Top:** Sea levels in Pärnu during the seiche between 5 - 9 October 2023 for 24 h (**left column**) and 72 h (**right column**) lead times. Orange area denotes the limits of the HIDRA2 ensemble envelope. **Bottom:** Bandpass filtered sea levels in the frequency band $(3h)^{-1} - (8h)^{-1}$ during the same period and for the same lead times. Note different vertical scales on both plots.

to some extent and successfully matches the seiche phase during periods of highest amplitudes. In contrast, the HIDRA2 ensemble mean is overly smooth, rendering it less capable of reproducing sea-level variability within the seiche frequency
band. However, it adheres more closely to the overall observations than NEMO$_{\mathrm{BAL}}$. HIDRA2's limited accuracy during seiche excitations leads to deviations from instantaneous sea-level values, increasing the short-term errors. However, over periods of several days, HIDRA2 exhibits minimal bias. This indicates that while individual hourly predictions may show higher error, HIDRA2's forecasts do not consistently over- or underestimate SSH over longer time spans, resulting in minimal systematic bias.

This experiment also demonstrates the value of the ensemble prediction. While separate members of the HIDRA2 ensemble (and consequently their ensemble mean as well) are not capable of capturing the high-frequency variability of the seiche (not shown), this limitation is partly compensated by the ensemble spread, which to some extent envelops the actual amplitude of the seiches. Therefore the sea levels during the seiche will often still be within the ensemble envelope although the seiche itself is poorly reproduced in each individual ensemble member. This ensemble approach in HIDRA2 is especially useful
for addressing the challenges of triggering events and managing uncertainty in sea level forecasts. Triggering events often involve abrupt and transient atmospheric or oceanic changes, like storms or pressure systems, which lead to rapid sea level shifts. While HIDRA2's ensemble mean may not directly capture these fluctuations, the ensemble spread creates a "buffer" around the observed values. This means that during trigger events, even if individual ensemble members miss exact timings or magnitudes, the ensemble as a whole may still envelop the observed extremes, providing a probabilistic signal of potential
risks.





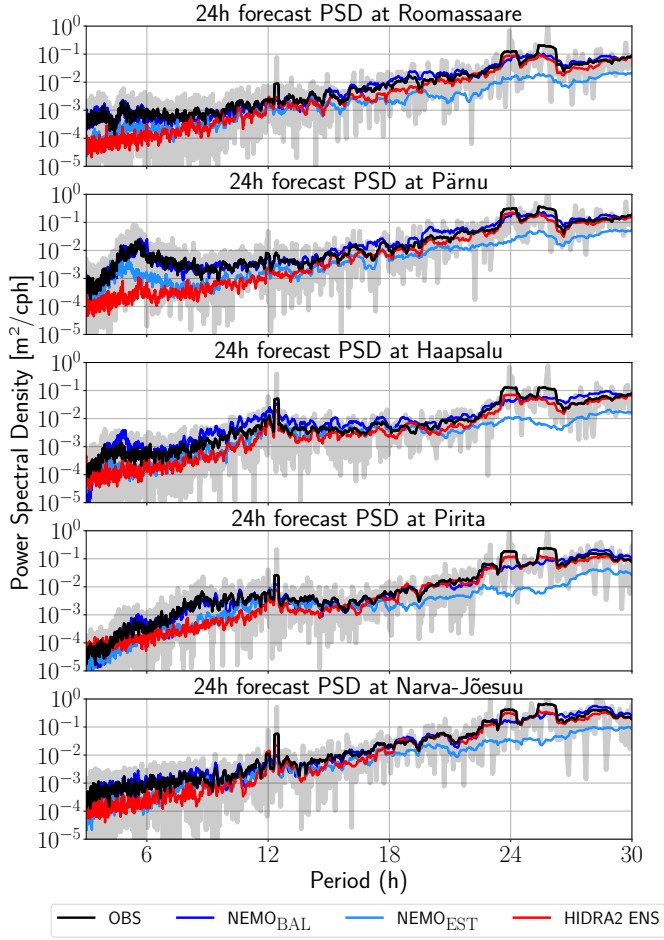

**Figure 9.** Power spectral densities [m$^2$/cph] of sea levels from different sources (colored lines, see legend at the bottom) at each station (rows). Denotation cph indicates cycles per hour.

To quantify NEMO$_{BAL}$, and HIDRA2 behavior in different spectral bands, we computed power spectral densities (PSD) for observations and model forecasts for each of the stations, as shown on Figure 9. PSD was calculated by normalizing the squared magnitude of the Fast Fourier Transform coefficients at each frequency, followed by an inversion of frequencies to periods. This allowed us to identify dominant seiche periods and the (ir)relevance of tidal forcing (peaks around 12 and 24 h). In Pärnu, for example, the fundamental seiche peak close to 6 h period is quite prominent, while the semi-diurnal 12 h peak does not stand out in its energy band, indicating that semi-diurnal tides are not driving the dynamics at this station. At other stations, for example, semi-diurnal tides are a bit more pronounced.

The spectra reveal that HIDRA2 faces challenges in fully capturing seiche dynamics and tends to underestimate energy density in the subinertial energy band (at the latitude of Pärnu, this corresponds to periods below 13 h or rather frequencies above $(13\,\text{h})^{-1}$ - in this band NEMO$_{BAL}$ accurately reproduces the observed amount of energy. HIDRA2 spectra are however





the closest to observations spectra in the band with periods above 18 h. Especially at diurnal tide band HIDRA2 shows slightly better performance than NEMO$_{BAL}$, while both HIDRA2 and NEMO$_{BAL}$ outperform NEMO$_{EST}$. This corresponds to the solid overall performance of HIDRA2 and its poor reproduction of high frequency variability. NEMO$_{EST}$ on the other hand lacks almost an order of magnitude of energy in the energy band with periods above 18 h, but indicates a decent reproduction of the
Pärnu seiche.

## 4    Discussion

While the hydrodynamic model NEMO$_{BAL}$ may be particularly more effective at detecting specific oscillations, such as the excited seiches, HIDRA2's adaptability makes it better suited for a wider range of sea level conditions. HIDRA2 consistently achieves lower RMSD values and frequently outperforms hydrodynamic models in predictive accuracy. This suggests that
the accuracy of hydrodynamic models in capturing short-term oscillations largely stems from their foundation in physical equations, including the use of temporal derivatives with high-resolution time-stepping, and their broad modeling domains, which enable responsive adaptation to changing environmental conditions across various scales. This approach suggests a deeper, more intrinsic understanding of the system's natural behavior. As a result, although data-driven models are adept at detecting data value fluctuations, they lack the inherent physical insight that hydrodynamic models possess (Zhao et al., 2024).
Nonetheless, future advancements in areas such as physics-informed deep learning can provide strong potential to enhance data-driven models for ocean parameter prediction, equipping them with a more nuanced, system-aware perspective (Raissi et al., 2019; Donnelly et al., 2024). However, developing an effective physics-informed deep learning model tailored to sea level prediction remains significantly more challenging than for other oceanic parameters, as sea level dynamics involve continuity and momentum equations that govern the conservation of mass and momentum under complex external influences.

An additional advantage of deep learning models, such as HIDRA2, lies in their computational efficiency. In the present study, HIDRA2 demonstrates the ability to generate 50 ensemble predictions for each 72-hour forecast at each station in approximately 30 seconds using a current typical personal computer or laptop (with performance rates typically ranging within the teraflops scale), using only atmospheric input data (wind and mean sea level pressure), background sea level, and the trained network file. In contrast, running a high-resolution hydrodynamic model for even a single ensemble simulation requires sig-
nificant computational resources, a large amount of input data, and considerable processing time, often necessitating dedicated high-performance computing facilities. Consequently, HIDRA2's efficiency not only enhances forecast accessibility but also enables a more adaptable and resource-effective approach to sea level prediction in operational settings, providing valuable support for optimizing processes in coastal management.

Furthermore, one inherent limitation of hydrodynamic models, particularly in accounting for long-term variability, lies in the
incorporation of a realistic vertical datum, which requires adjustments for dynamic geoid height variations and land movements (Jahanmard et al., 2021; Mostafavi et al., 2023). These adjustments pose additional challenges to the accuracy of traditional models, which rely on stable reference levels and may struggle to account for local geophysical changes (Jahanmard et al., 2022). This constraint further underscores the advantage of deep learning models like HIDRA2, which do not rely on prede-



fined or geophysically biased dynamic topography but instead adapt more flexibly to complex patterns within large datasets,
capturing broad behaviors and fine details with high fidelity.

Besides, validation metrics, such as RMSD and correlation coefficient, presented in this study underscore HIDRA2's superior performance compared to other sea level forecasting methods, as demonstrated in studies e.g. Ishida et al. (2020); Shahabi and Tahvildari (2024); Dong et al. (2024). Despite the influence of various regional factors on short-term SSH forecasts, the primary distinction between HIDRA2 and methods used in previous studies is the utilization of a deep-learning "ensemble" approach. Thus, the effectiveness of HIDRA2 in this case study underscores the importance of incorporating diverse ensembles to account for varying short-term atmospheric conditions and climate scenarios (Gröger et al., 2019; Hieronymus and Hieronymus, 2023). For operational applications, particularly in decision-making and initiating specific response phases, HIDRA2 can be more effective when the triggering limits are set to the maximum and minimum ensemble values, as these represent the most probable scenarios. Although the probability of reaching the maximum or minimum ensemble values in typical cases is around 2%—given the 50 ensembles used in this study—the ensemble spread, which almost always encompasses the absolute SSH values, suggests that HIDRA2 is a more reliable tool for operational forecasting and triggering. This reliability is especially notable when compared to other forecasting systems that do not provide ensemble predictions. Future studies could further explore optimizing this ensemble range. In the present study, it has been shown that the spread of ensemble-predicted values increases with longer lead times, with this spread becoming particularly pronounced during extreme observed SSH values. When considering a separate evaluation of negative and positive extreme SSH events, the results show that for negative extreme SSH events, the correlation is lower with a narrower RMSD, while for positive extremes, the correlation is higher but accompanied by a wider RMSD. Here, the significance of comparing RMSDs may be less relevant, as higher RMSD values can be acceptable when target values are higher. But, in general, the lower correlation during negative SSH extremes suggests that while the model captures some general trends or patterns, it does not align well with the exact fluctuations or magnitudes of observed values. From a data-driven modeling perspective, this may be explained by data imbalance: negative extreme events are rarer than positive extremes in the study area (Conte and Lionello, 2013; Jensen et al., 2022), providing HIDRA2 with fewer examples of negative events during training. Consequently, HIDRA2 may perform better for positive extreme events, having learned from their relatively higher frequency. This observation demonstrates the model's enhanced capability to capture more common extreme event types.

## 5 Conclusions

This study evaluates an implementation of HIDRA2, a deep-learning sea level model, alongside two numerical models, NEMO$_{EST}$, a subregional hydrodynamic model with a spatial resolution of 0.5 nautical miles, and NEMO$_{BAL}$, a regional hydrodynamic model with a spatial resolution of 1 nautical mile and a 12-hour lead time. The study area focuses on five tide gauge stations along the Estonian coast of the Baltic Sea: Narva-Jõesuu, Pritia, Haapsalu, Pärnu, and Roomasaare, which provided hourly SSH observations for the study period from April 1, 2023, to March 31, 2024. In summary, based on the



averaged correlation coefficient and RMSD across the study period between modeled and observed SSH, the results indicate that HIDRA2 generally outperforms the hydrodynamic models.

While accuracy decreases in predicting extreme sea level events, the observed values generally fall within the predicted ensemble spread of HIDRA2, highlighting its overall reliability. However, some open issues remain. HIDRA2 did not perfectly reproduce seiches in Pärnu bay and in general lacks energy in the subinertial band with periods below 13 h. Reproduction of sea level dynamics in this band is something that will have to be addressed in further work. This can be partly offset by an ensemble spread but the spread with 24 h lead time is underestimated while the spread for 72 h seems overestimated, as can be seen from Figure 8. This remains a challenge for future investigations.

Nevertheless, by utilizing HIDRA2's ensemble spread, stakeholders can gain greater confidence in the forecast range rather than relying solely on a deterministic forecast. For example, if the ensemble spread widens around an event, it indicates a higher level of uncertainty and suggests preparing for a range of scenarios, rather than a single outcome. Ensemble models like HIDRA2 can be set to generate alerts when the spread surpasses a specific threshold, signaling a high probability of an unusual event. This probabilistic signaling can be essential for coastal management, providing early warnings for potential risks in the face of triggering events, even if the ensemble mean remains stable.

Deep-learning ensemble models, such as HIDRA2, are pertinent for advancing the development of Digital Twins and associated impact models (Li et al., 2023). These models, utilizing ensemble-based techniques, are particularly effective in capturing the complex, non-linear relationships in SSH data across diverse scales. By integrating multiple predictive models, ensemble approaches enhance the accuracy and robustness of forecasts, making them valuable for the creation of Digital Twins of the Earth systems. This, in turn, supports more precise impact assessments and decision-making processes in coastal management and risk mitigation.

*Code availability.* The persistent version of the GitHub repository containing HIDRA2 code is available at https://doi.org/10.5281/zenodo.7307365.

*Data availability.* SSH observation data for each studied station are available from the Estonian Environmental Agency (https://www.ilmateenistus.ee/). Atmospheric inputs used for training and running the models in this study can be obtained from ECMWF (https://www.ecmwf.int/en/forecasts/). Baltic Sea Physics Analysis and Forecast datasets (referred to as NEMO$_{BAL}$ in this study), are accessible through the EU Copernicus Marine Service under the dataset identifier BALTICSEA_ANALYSISFORECAST_PHY_003_006 (http://marine.copernicus.eu/).

*Author contributions.* All authors contributed to discussions and editing of the paper. **A.B.** conducted model runs and data extractions for the study area, pre- and post-processed the outputs, analyzed and prepared the final results, and drafted the original manuscript. **M.R.** prepared and trained the HIDRA2 model for the case study. **M.L.** performed spectral analyses and wrote parts of the manuscript. **A.B.** and **M.L.**



plotted the Figures. **M.L.**, **I.M.**, and **R.U.** contributed to the design and conceptualization of the study and provided advisory input on the methodology. **I.M.**, **J.E.**, and **P.L.** conducted the NEMO$_{EST}$ runs and prepared and preprocessed the related data.

*Competing interests.* The authors declare that they have no conflict of interest.

*Financial support.* This research was funded by the EU through agreement DE_330_MF between ECMWF and Météo-France. Additional support was provided by the EU and the Estonian Research Council under project TEM-TA38 (Digital Twin of Marine Renewable Energy). Furthermore, this work received funding through the AdapEST project ("Implementation of national climate change adaptation activities in Estonia, VEU23019"), supported by the European Climate, Infrastructure and Environment Executive Agency (CINEA) via the LIFE programme. **M.L.** acknowledges the financial support from the Slovenian Research and Innovation Agency ARIS (contract no. P1-0237).



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
