# Peer review of "Application of HIDRA2 Deep Learning Model for Sea Level Forecasting Along the Estonian Coast of the Baltic Sea"

_EGUsphere, 2024_

## Author Comment (AC1)

Dear Reviewer, we appreciate the valuable comments and the time you dedicated to reviewing our manuscript. In the following, we provide detailed responses to each comment (purple notes). In addition we revised the manuscript accordingly in this regard (green notes).

**1st Reviewer:**

**GENERAL COMMENTS**

This study presents the main results obtained from the application of the HIDRA2 deep learning model along the Estonian coast for sea level prediction. It compares the model's performance with hydrodynamic models and provides an analysis of the performance of these models based on different components of sea level.

Overall, the document is well-structured, including a clear description of the data and methods used, a detailed presentation of the main results, a discussion of significant limitations of the applied model, and a conclusion summarizing the key findings.

However, there are several important points that the authors should address, as outlined below:

**SPECIFIC COMMENTS**

Section 2.1: Could you please provide the time coverage of the data at each location? Hourly SSH observations have been available since 2010 for all five stations, coinciding with the initiation of routine validation/control measurements conducted by the Estonian Environmental Agency. In this study, the dataset covering 2010 to 2019 was utilized for training the HIDRA2 model. Data from 2020 to March 2023 were checked for additional tuning to enhance the robustness of the HIDRA2 application developed for this study. Finally, the period from April 2023 to April 2024 was reserved for independent testing and analysis presented in the current manuscript. For improved clarity and consistency, further modifications have been made in the revised manuscript text:

Line 78-79: The datasets were provided by the Estonian Environmental Agency (https://www.ilmateenistus.ee) and span the period between 15 June 2010 and 30 April 2024.

Line 106-108: However, the SSH data used for training consisted of hourly observations from 2010 to 2019 at all locations, while the period from April 2023 to April 2024 was utilized for testing the model and performing the analyses presented in this study.

Lines 97–98: "covering a longitudinal range from 16.25°E to 28.5°E and a latitudinal range from 54.25°N to 64°N". Is there a specific reason for the chosen spatial extent of the fields? Was it determined through trial and error during the study or based on a reference?
The domain of meteorological forcing excludes remote shallow areas west of the Bornholm Basin, the northernmost Bay of Bothnia, and the narrow Neva Bight in the eastern part of the Gulf of Finland. Covering longitudes from 16.25°E and

latitudes from 54.25°N to 64°N, the spatial extent was selected through iterative testing to minimize forecast errors while maintaining a suitable lead time. This configuration ensures the adequate capture and resolution of mesoscale meteorological systems relevant to the study region. A smaller domain was found to limit the visibility of approaching weather patterns, thereby reducing predictive capability. The chosen boundaries thus provide a balanced compromise between computational efficiency and accurate representation of key atmospheric processes affecting sea surface height dynamics.

For improved clarity and consistency, further modifications have been made in the revised manuscript text:

Line 98-99: This selection was guided by iterative testing aimed at minimizing forecast errors while ensuring a suitable lead time.

Lines 99–100: "The training data for SSH at the coastal stations were obtained from the Estonian Environmental Agency". Is this the same dataset mentioned in Section 2.1?

Yes, the dataset mentioned in Section 2.1 is the same. All SSH observation data used in this study—including those for training, tuning, and testing—were obtained from the Estonian Environmental Agency. We believe that the modifications outlined above (prior comments) improve clarity in this regard as well.

Line 101: "…the period 2010 to 2019". Does this timeframe apply to all five locations? How much of the data was used for training versus testing the model? Did you analyze the data for trends? If so, were they removed, or is the model capable of accounting for them?

Yes, the timeframe from 2010 to 2019 applies to all five locations, and this period was used exclusively for training the HIDRA2 model. The testing time is from April 2023 to March 2024, as it is already clarified in the text.

For trends, we did not explicitly analyze or remove long-term trends from the data. The reason is that while linear trend analysis is a common approach in traditional time series studies, it is relatively low-dimensional and easily captured by deep learning models such as HIDRA2. Therefore, we relied on the model's capability to account for such patterns internally during training. However, elementary quality control was performed on the SSH data to identify and handle missing values and anomalous observations, ensuring clean and consistent inputs for both the training and inference phases.

Line 107: What do you mean by "adapted"? Did the BALMFC apply modifications to the model? Or do you mean "implemented"?

The intended meaning was that BALMFC implemented the referenced model and its configuration for operational forecasting in the Baltic Sea. The manuscript is revised accordingly.

Lines 155–162: The comments regarding 24-hour and 72-hour lead times should focus on the comparison between HIDRA2 and NEMO$_{EST}$, as Table 1 provides results for only the 12-hour lead time for NEMO$_{BAL}$. Am I interpreting this correctly? I suggest reorganizing this paragraph slightly to make the description of results more fluid for the reader.

Yes, your interpretation is correct; we have revised the paragraph as follows:

Line 161: Table 1 presents the RMSD values for the forecasts across all tide gauge stations, corresponding to the availability of model outputs in the present study.

Additionally, while reading the document, I expected a comparison between HIDRA2 and NEMO$_{EST}$, as the latter is expected to have better performance compared to NEMO$_{BAL}$. This assumption arises because NEMO$_{EST}$ has a higher resolution and uses NEMO$_{BAL}$ outputs as boundary conditions, which could lead to improved performance. Did you anticipate this performance hierarchy for NEMO$_{EST}$? It may be beneficial to provide a more explicit description of the relative performance of all three models in this section.

We consider your expectation noteworthy that NEMO$_{EST}$, due to its higher-resolution grid, might outperform NEMO$_{BAL}$. However, both models were run independently; NEMO$_{EST}$ is not nested within NEMO$_{BAL}$, although its boundary conditions were derived from NEMO$_{BAL}$ outputs at each time step.

For further clarification, we should note that while higher-resolution models can better capture fine-scale features such as intricate coastlines and bathymetric variations, the extent of the model domain also plays a pivotal role, especially in regions affected by large-scale dynamics. For example, the larger spatial domain in NEMOBAL allows for the evolution of basin-wide seiches (e.g., Baltic Sea seiches), which can propagate into sub-regions such as the Gulf of Riga or the Gulf of Finland. These basin-wide dynamics can interact with sub-regional dynamics, potentially affecting local forecast accuracy.

To maintain the focus of our study, we did not conduct a comprehensive comparative analysis between NEMO$_{BAL}$ and NEMO$_{EST}$. Nonetheless, we acknowledge the importance of considering both model resolution and domain size when evaluating performance, as each factor contributes uniquely to the accuracy of sea level forecasts in complex coastal regions.

Lines 155–156: "The RMSDs for each station are calculated using sea level data from April 2023 to April 2024". I suggest moving this statement to the Methods section or the section where validation metrics are introduced.

We have modified and moved the sentence to the end of the Methods section, where the validation metrics are introduced, to improve clarity and logical flow. The revised sentence now reads:

Line 157: The validation metrics for each station are calculated using referenced to the SSH observation data from April 2023 to April 2024.

Lines 164–167: "The performances are also separately assessed during extreme negative and positive observed SSH events (Figure 4b & c). Extreme negative SSH values are defined as those falling below the 5th percentile of observed SSH during the study period at each station, while extreme positive values exceed the 95th percentile (Cannaby et al., 2016; Mentaschi et al., 2023)". I recommend moving this explanation to the Methods section.

We have modified and moved the sentence to the end of the "Forecasting and evaluation framework" subsection, where the validation metrics are introduced, to improve clarity and logical flow. The revised sentence now reads:

Line 142-145: Furthermore, the performance of the forecasts is also separately evaluated during extreme negative and positive observed SSH events. Extreme negative SSH values are defined as those falling below the 5th percentile of observed SSH at each station during the study period, while extreme positive values are those exceeding the 95th percentile (Cannaby et al., 2016; Mentaschi et al., 2023).

Lines 177–178: "For negative extreme SSH events, correlation is lower with a lower RMSD, while for positive extreme SSH events, correlation is higher but with a higher RMSD". These comments appear to refer to Figure 5. If this is correct, please explicitly indicate this in the text. Additionally, I suggest including this description before presenting the figure.

You are correct — the statement refers to the results shown in **Figure 5**. We have revised the manuscript to explicitly reference Figure 5 in this context and have moved the description to precede the figure to improve clarity and logical flow.

Lines 181–182: "For clarity, the performance of NEMO$_{EST}$, which is the least accurate (as detailed in Section 3.1), is not included in the figure". The only explicit statement supporting this claim is found in Lines 171–172: "while the subregional model NEMO$_{EST}$ performs better under non-extreme high SSH conditions but struggles to accurately predict extreme SSH values". This could be clarified further.

We acknowledge that the explanation in the text could be more explicitly linked to the quantitative results. The statement in Lines 181–182 was specifically supported by the performance metrics presented in **Table 1** within Section 3.1, which reports RMSD values for each forecast model across multiple lead times. To improve clarity, we have revised the manuscript to explicitly reference **Table 1** rather than broadly citing Section 3.1. Additionally, in response to a suggestion from another reviewer, we have moved Figure 6 and its notes to the supplementary material.

Line 182: What do you mean by "stable behavior"? Are you referring to the absence of gaps or offsets in the models, or to the models' ability to consistently replicate observed time series?

In this context, we are referring to the models' ability to consistently follow the fluctuation patterns observed in the time series, even under extreme conditions. In the revised version of the manuscript, the term "stable behavior" has been removed.

Line 182: I have some doubts regarding the use of the term "timescales". Are you referring to the temporal range used for this comparison?

In this context, our intention was to refer to the **temporal range** over which the comparison between forecasts and observations was made. To avoid ambiguity, we removed this excessive part.

Line 187: "A visual comparison of model predictions on Figure 7 between 5 and 9 October 2023". This part is a bit confusing. The seiche representation is not clearly visible in Figure 7, so I initially assumed the reference was to Figure 8. However, based on subsequent context, it seems these comments about the seiche do refer to Figure 7. Could you please confirm this? If the seiche is depicted in Figure 7, I suggest clearly highlighting it in the figure to avoid confusion.

You are correct — the discussion referred to **Figure 7** (now it has been renamed to Figure 6 because we moved a prior figure to the supplementary material), where rapid fluctuations observed between 5 and 9 October 2023 in Pärnu Bay are visually recognized as a potential seiche event. This interpretation is based on previously documented seiche behavior in the bay, as described in Lines 62–67 of the Introduction. These fluctuations appear as high-frequency, oscillation-like patterns in the observational time series (black line), whereas the HIDRA2 prediction (red line) exhibits a more smoothed response.

As already explained in the manuscript, this observation motivated the additional analysis presented in the next figure (Figure 7 in the revised manuscript). However, to improve clarity for readers, we have revised Figure 6 to include an orange box highlighting the relevant period (5–9 October 2023), which serves as the basis for the deeper investigation shown in Figure 7.

Lines 198–204: "In contrast, the HIDRA2 ensemble mean is overly smooth, rendering it less capable of reproducing sea-level variability within the seiche frequency band. However, it adheres more closely to the overall observations than NEMOBAL. HIDRA2's limited accuracy during seiche excitations leads to deviations from instantaneous sea-level values, increasing the short-term errors. However, over periods of several days, HIDRA2 exhibits minimal bias. This indicates that while individual hourly predictions may show higher error, HIDRA2's forecasts do not consistently over- or underestimate SSH over longer time spans, resulting in minimal systematic bias".
From Table 1, it is evident that HIDRA2 achieves a better RMSD compared to NEMOBAL at Pärnu. Given that NEMO$_{BAL}$ appears to attempt capturing the seiche and the highest sea levels, did you consider whether the RMSD "double penalty" might have affected NEMO$_{BAL}$'s evaluation, leading to HIDRA2 achieving a better score?

A double penalty is indeed often a problem in an evaluation. When speaking of storm surges, the double penalty is typically related to a temporal mismatch between modeled seiche and observed seiche. This mismatch does, however, appropriately reflect an actual double trouble: it reflects the fact that high SSH was predicted when

low water occurred, and that low SSH was predicted when high water occurred. Both mistakes have consequences: high SSH means flooding, and low SSH may impede marine traffic. Therefore a double penalty is not a completely unfair penalty in this particular situation. Furthermore, Figure 7 shows that RMSD also reflects a general SSH overestimation of NEMO$_{BAL}$ and is not only related to seiche reproduction. Such overestimation will not lead to any double penalty. HIDRA2, on the other hand, exhibits much lower bias. Further insight into error behaviour can be seen using spectral analysis, shown in Figure 8. It is clear from Figure 8 that HIDRA2 does not reproduce seiches with a ~ 6h period, but it is worth noting that NEMO$_{BAL}$ is systematically underestimating processes with periods over 18h. And these processes contain up to 100 times more energy since the scale on Figure 9 is logarithmic. In other words, HIDRA2 is reproducing the more energetic part of the spectra with higher precision, but fails to reproduce seiches which contain 10-100 times less energy. In this sense, lower RMSD from HIDRA2 is not an artificial metric that stems from an unfair penalty to the NEMO$_{BAL}$.

We have, however, incorporated reviewers' concerns into a new paragraph where we pointed out the possibility of a double penalty and our comments about it (lines 166-174).

I suggest incorporating additional performance statistics beyond RMSD to provide a more robust evaluation of the models. While RMSD offers a general sense of performance, it may not fully account for specific limitations, such as the "double penalty" effect. The following references may prove useful for clarifying the results obtained in your study:

- https://doi.org/10.1016/j.ocemod.2013.08.003
- https://doi.org/10.5194/os-20-1513-2024

We thank the reviewer for these references. The second reference (Campos-Bala et al., Ocean Science 2024) suggests that a mean absolute deviation, corrected with percentile deviations (called MADc = MAD + MADp) might be a more suitable metric for model performance. We agree. In this context, we would like to note that while Table 1 does indeed only list RMSD, we plot RMSD over all percentiles in Figure 3. Figure 3 therefore effectively already graphically depicts what MADc measures: error by sea level bin / percentile. We can see that HIDRA2 (red line in Figure 3) has the lower error over (practically) the entire SSH range, i.e. at all percentiles.

We have therefore expanded the paragraph about Table 1 to reflect reviewers' comments about weaknesses of RMSD, and we also included the reference to (Campos-Bala et al., 2024) along with an observation that HIDRA2 performs better than NEMO in all SSH bins. The paragraph now reads:

Several studies such as Campos-Caba et al. (2024) and Mentaschi et al. (2013) have highlighted limitations in using the RMSD as a standalone performance metric, particularly due to the double penalty effect. This occurs when temporal mismatches between modeled and observed seiches lead to models being penalized twice—once for predicting a peak that did not occur, and again for failing to predict one that did. As a result, RMSD may not fully capture the quality of model predictions.

170 To address this, RMSD should be interpreted alongside complementary performance metrics. For instance, Campos-Caba et al. (2024) propose evaluating the mean absolute deviation $\overline{|model - observation|}$, adjusted across percentiles by the corresponding mean absolute deviation $\overline{|model_p - observation_p|}$ from the 0th to the 100th percentile in 1% steps.

We therefore recommend that Table 1 and the RMSD values be considered in conjunction with the model error across all sea level bins (Figure 3) and further contextualized through spectral analysis of model and observed time series (Figure 8).

**TECHNICAL CORRECTIONS**

Figure 1: There is an error in the y-axis label of Figure 1. It should read "Latitude" instead of "Longitude."

This correction has been implemented in the revised manuscript.

Line 175: Replace "predeicted" with "predicted."

This correction has been implemented in the revised manuscript.

Figure 7: I suggest including a proper title for Figure 7, even if most of the title overlaps with Figure 6. This ensures the reader does not need to refer back to the previous figure to understand the context of Figure 7.

This modification has been implemented in the revised manuscript.

Line 187: The phrase "quite telling" appears somewhat informal for a scientific article. A more suitable synonym could be "highly indicative" or "clearly demonstrates."

This correction has been implemented in the revised manuscript.

Lines 189 to 192: The sentence, "We have high-frequency variability in NEMO$_{BAL}$ and a completely smooth HIDRA2 ensemble mean. One might argue that this smoothing stems from the fact that we are working with the ensemble mean in which otherwise present oscillations in different ensemble members cancel out," could be improved by rephrasing to avoid informal language. Suggested revision:

"NEMO$_{BAL}$ exhibits high-frequency variability, whereas the HIDRA2 ensemble mean is completely smooth. This smoothing effect may result from averaging the ensemble members, where oscillations present in individual members cancel each other out."

The suggestion has been implemented in the revised manuscript.

Lines 196 to 197: The sentence, "A clear excitation of the seiche is visible in filtered observations in Figure 8 from 3 October on NEMO$_{BAL}$ captures this excitation," could be rephrased for clarity as: "Filtered observations in Figure 8 clearly show the excitation of the seiche starting on 3 October, which is also captured by NEMO$_{BAL}$."

The suggestion has been implemented in the revised manuscript.

Figure 9: I recommend placing Figure 9 immediately after the text where it is mentioned for the first time. This improves readability and ensures the reader can quickly reference the figure without searching for it later in the document.

This modification has been implemented in the revised manuscript.

---

## Author Comment (AC2)

**2nd Reviewer:**

**GENERAL COMMENTS**

This study presents the results of applying the HIDRA2 deep learning model to forecast sea level along the Estonian coast of the Baltic Sea. Similar to its application in the Adriatic Sea, for which HIDRA2 was originally developed, the data-driven model generally outperforms dynamical models (3D ocean models), except for extreme events. This is expected, as extreme events are inherently rare and, therefore, challenging to accurately represent using a data-driven approach.

While the manuscript is clear and well-written, it does not significantly advance the understanding of sea surface height (SSH) forecasting or machine learning (ML)-based methods for this purpose. However, it provides a valuable report on a state-of-the-art ML-based system capable of producing fast and computationally "cheap" SSH forecasts for selected locations within the Baltic Sea.

I strongly encourage the authors to reduce the length of certain sections, particularly in the discussion and conclusion (which often reads like a summary), where some paragraphs are repetitive or restate well-known concepts—such as the efficiency and computational cost-effectiveness of ML methods compared to 3D ocean models based on primitive equations. Instead, I suggest expanding on the ensemble approach, assessing its limitations, and exploring potential strategies to improve the representativenes of the ensemble spread.

We thank the reviewer for encouraging and constructive remarks. We will follow their suggestion and significantly shortened suggested sections.

Other comments:

Line 89: Brackets should be only around the year: "Details of the encoding architecture are presented in (Rus et al. 2023)."

Thank you, corrected.

Lines 98-99: "The original meteorological data, with a domain size of 40 × 50, were subsampled to a 9 × 12 grid."

Subsampling appears to discard valuable information. Have the authors attempted to use the full resolution? Do you anticipate any improvements by retaining the original grid size?

The reviewer is correct to point this out. This does merit further clarification. We did indeed conduct experiments using the full-resolution atmospheric fields. These trials involved necessary adjustments and finetuning of the model. However, we found that the resulting performance metrics were comparable to those achieved with the subsampled data, showing no significant improvement despite the increased computational demand. Based on these empirical results, while the intuition that higher resolution might hold more information is valid, our current model configuration did not seem to benefit from it for this specific task. Given the similar performance and the considerable advantage in computational efficiency, we retained the subsampled approach for the results presented.

In our implementation, the original 40 × 50 meteorological fields were not simply subsampled but downscaled to a 9 × 12 grid using **bilinear interpolation** (PyTorch's => Resize function). This approach retains the large-scale spatial patterns while significantly reducing the input dimensionality, allowing for efficient model training and inference.

We did experiment with retaining the full-resolution input during early testing. However, the increased computational cost did not yield meaningful improvements in predictive performance, particularly in metrics such as RMSD and correlation. Given this trade-off, the interpolated grid was selected to balance model complexity and skill.

We have clarified this in the revised manuscript to indicate that bilinear interpolation was used and that the grid reduction was a design choice informed by iterative testing. Also, we acknowledge that the manuscript previously used the term **"linear interpolation"**, which has now been corrected to **"bilinear interpolation"** to accurately reflect the 2D nature of the operation applied to the spatial grid.

we have revised the manuscript as follows:

to 28.5°E and a latitudinal range from 54.25°N to 64°N. This selection was guided by iterative testing aimed at minimizing forecast errors. The original meteorological data, with a domain size of $40 \times 50$, were subsampled to a $9 \times 12$ grid using bilinear 100 interpolation to match HIDRA2's required input size. This transformation reduced dimensionality while preserving key spatial patterns, facilitating more efficient model training. We also conducted experiments using the full-resolution atmospheric fields. These trials involved necessary adjustments and finetuning of the model. However, we found that the resulting performance

metrics were comparable to those achieved with the subsampled data, showing no significant improvement despite the increased computational demand. Given the similar performance and the considerable advantage in computational efficiency, we retained 105 the subsampled approach for the results presented. The training data for SSH at the coastal stations were obtained from the

Figure 2: It is misleading to depict the model architecture using the Adriatic Sea instead of the Baltic Sea. If this figure is sourced from another article, please provide the appropriate citation. If it was created specifically for this study, consider replacing the Adriatic Sea with the Baltic Sea to avoid confusion.

We agree. The Figure was now changed to feature the Baltic domain.

Figure 6: This figure is not particularly informative, as most curves overlap, except for Feb-Mar 2024 in Haapsalu. Consider moving it to the supplementary material, as the key point is already well illustrated in Figure 7.

suggestion accepted. Figure 6 was moved to the supplementary material.

Line 190: "One might argue that this smoothing stems from the fact that we are working with the ensemble mean". Please include a figure like Figure 7, but for both the best and worst ensemble members (perhaps based on RMSE). This will help illustrate the smoothing effect of the ensemble mean more effectively.

It should be noted that, in principle, no individual ensemble member performs consistently better or worse than the others. In the ECMWF Ensemble Prediction System (EPS), all members are reinitialized at each forecast cycle. This means, for example, that the 7th member in today's forecast is unrelated to the 7th member from yesterday or tomorrow. There is no continuity between members over time—each one is statistically equivalent in terms of performance. Therefore, plotting the best and worst members across multiple cycles would result in a large number of discontinuous curves, which would reduce the clarity of the figure. We hope the reviewer agrees that such a representation would not be helpful. Nevertheless, the following clarification has been added to the Figure caption: "Grey lines in the background denote raw spectra, while black lines denote their 6-hour moving average."

Lines 230-250: The discussion in this section largely reiterates well-established points about HYDRA2 without adding new insights. Writing a full paragraph to restate that ML-based methods are significantly more computationally efficient than traditional dynamical models seem unnecessary, as this fact is widely recognized, even by non-specialists.

This section has been significantly shortened and mentioned paragraphs were removed in the interest of brevity and non-repetitiveness.

Line 283: This observation demonstrates the model's enhanced capability to capture more common extreme event types. This idea should be discussed also for prediction of normal vs extremes (rare) SSH conditions, where HIDRA2 generally outperforms dynamical models.

In the revised manuscript the discussion has been extended as follow:

extreme events, having learned from their relatively higher frequency. This observation demonstrates the model's enhanced capability to capture more common extreme event types. This stems from HIDRA2's data-driven foundation, which allows it to effectively learn patterns from densely populated regions of the training distribution. In contrast, hydrodynamic models, while grounded in first-principles physics, are generally less responsive to data frequency and often lack flexibility in proba- bilistic forecasting. To further improve HIDRA2's performance across the full range of sea level conditions—particularly in data-sparse regimes—future developments could explore the integration of physics-informed neural networks (Raissi et al., 2019; Zhu et al., 2025). By incorporating physical constraints, such as conservation laws or shallow water dynamics, into the learning process, such approaches can mitigate the effects of limited data availability while guiding the model toward physi- cally consistent behavior, even under rare or extreme scenarios. Looking ahead, combining physics-informed strategies with ensemble-based deep learning may provide a more robust and generalizable framework for sea level forecasting, supporting both routine operations and high-impact coastal applications.

Line 286-292: This section reads more like a summary rather than a conclusion and should be revised accordingly.

This section was thoroughly shortened and substantially rewritten along the lines of the reviewer's suggestions.

Line 305: I do not find this section informative. Deep-learning ensemble models, such as HIDRA2, are pertinent for advancing the development of Digital Twins and associated impact models (Li et al., 2023). These models, utilizing ensemble-based techniques, are particularly effective in capturing the complex, non-linear relationships in SSH data across diverse scales. By integrating multiple predictive models, ensemble approaches enhance the accuracy and robustness of forecasts, making them valuable for the creation of Digital Twins of the Earth systems. This, in turn, supports more precise impact assessments and decision-making processes in coastal management and risk mitigation.

We agree. This section was removed from the paper.